# Associations Between Carbohydrate Intake Behaviours and Glycaemia in Gestational Diabetes: A Prospective Observational Study

**DOI:** 10.3390/nu17030400

**Published:** 2025-01-22

**Authors:** Roslyn Smith, Renee Borg, Vincent Wong, Hamish Russell, Ka Hi Mak

**Affiliations:** 1Diabetes and Endocrine Service, Liverpool Hospital, Sydney, NSW 2170, Australia; 2Department of Dietetics, Liverpool Hospital, Sydney, NSW 2170, Australia; 3Hammondcare Braeside Hospital, Prairiewood, Sydney, NSW 2176, Australia; 4School of Clinical Medicine, University of New South Wales, South West Sydney Clinical Campus, Liverpool, NSW 2170, Australia

**Keywords:** gestational diabetes, dietary carbohydrates, dietary behaviour, blood glucose self monitoring

## Abstract

Carbohydrate intake (CI) has the largest impact on the short-term glycaemia of all nutrients, yet optimal CI management in gestational diabetes remains unclear. Objective: To examine CI behaviours in individuals with recently diagnosed gestational diabetes and their association with self-monitored blood glucose. Methods: Data from 97 individuals were collected using food and blood glucose diaries. CI (including amounts, types, and timing) was manually assessed using 15 g servings over 5–8 days, while a 3-day computerised analysis examined a broader intake of macro- and micronutrients. Results: Elevated fasting glycaemia (EFG) was associated with lower total CI (Mdn 10.8 vs. 12.5 servings/day, *p* = 0.006), missed meals (Mdn 1.4 vs. 0.0/week, *p* = 0.007), missed snacks (Mdn 10.5 vs. 7.5/week, *p* = 0.038), low-carbohydrate meals (<30 g; Mdn 4.3 vs. 2.0/week, *p* = 0.004), and a higher proportion of energy intake from fat (Mdn 35% vs. 33%, *p* = 0.047), as compared with in-target fasting glycaemia. In contrast, elevated postprandial glycaemia (EPG) was not significantly associated with total CI, low-carbohydrate meals (<30 g), low-carbohydrate snacks (<15 g), or high-carbohydrate loads (>50 g). EPG was instead associated with high-glycaemic index meals (Mdn 1.6 vs. 0.9 lunch meals/week, *p* = 0.026; Mdn 0.9 vs. 0.0 dinner meals/week, *p* = 0.023); and a lower intake of energy (Mdn 7650 kJ vs. 9070 kJ/day, *p* = 0.031), protein (Mdn 91 g vs. 109 g/day, *p* = 0.015), fat (Mdn 61 g vs. 84 g/day, *p* = 0.003), and multiple micronutrients (*p* <0.05), as compared with in-target postprandial glycaemia. Conclusions: CI profiles differed for individuals with EFG, as compared with EPG, emphasising the need for dietary guidelines tailored for gestational diabetes subtypes. Further interventional studies are needed to explore these findings, particularly the associations between EFG and low CI behaviours.

## 1. Introduction

Dietary intervention is widely recognised as the cornerstone of gestational diabetes management and has been associated with reductions in pregnancy complications, including low birthweight, improved haemoglobin A1c levels, and a decreased risk of macrosomia [1,2]. However, an optimal dietary approach remains undefined. Systematic reviews with meta-analyses have been unable to identify a superior dietary strategy, primarily due to the limited sample sizes of individual studies, which hinder robust comparison [2,3]. As a result, clinical practice guidelines exhibit variability in their nutritional recommendations, most of which are based on low-quality evidence [4].

Dietary carbohydrate in particular, is the nutrient that has the greatest impact on short-term glycaemia. However, optimal recommendations for carbohydrate intake (CI) remain unclear.

Earlier research suggested that postprandial glycaemic control required a lower proportion of energy from carbohydrates (30–40%) [5]. However, such diets are proportionally higher in dietary fat, which may raise circulating free fatty acids, promote inflammation, worsen insulin resistance, and contribute to foetal overgrowth [6]. More recent research indicates that glycaemic control can be achieved with higher intakes of nutrient-dense carbohydrates, which may enhance insulin action and have positive vascular effects [7]. Furthermore, supporting normal foetal growth while maintaining glycaemic control likely requires a CI between 47 and 70% of total energy [8].

In addition to total CI, types and sources of carbohydrates play a significant role. Low-glycaemic index (GI) carbohydrates, which are absorbed more slowly into the bloodstream, and higher intakes of dietary fibre help reduce postprandial glucose excursions and improve overall glycaemia [9,10]. Notably, glycaemic load (GL)—a measure that considers both the quantity of carbohydrates and their GI—may have the most substantial impact on outcomes. A recent systematic review on carbohydrate quality and quantity found that a low-GL diet may reduce the risk of macrosomia, while outcomes associated with low-GI and low-carbohydrate diets were comparable to those of usual care [11].

Regarding CI timing and distribution in gestational diabetes, a 2021 review identified only a limited number of primarily observational studies. The findings suggested that consuming more energy and carbohydrates earlier in the day, along with longer overnight fasting durations, may improve glycaemia [12]. Additionally, a randomised crossover study using continuous glucose monitoring reported lower mean glucose levels and reduced insulin resistance with a higher CI in the morning compared to the evening; however, this was accompanied by increased glucose variability [13].

Currently, the Academy of Nutrition and Dietetics recommends that CI guidance be individualised based on clinical assessment, consumer goals, and ongoing monitoring [14], while the American Diabetes Association advises a minimum CI of 175 g per day, with an emphasis on low-glycaemic index choices [15]. However, given the gaps in evidence-based knowledge, there are inconsistencies in dietetic practice. A survey of Australian dietitians identified marked differences in recommendations regarding minimum daily CI and percentage of energy derived from carbohydrates [16]. The study did identify some commonly employed strategies, including the distribution of CI across three main meals (each containing 30–45 g of carbohydrates) and two to three snacks (each containing 15–30 g of carbohydrates).

Research into the implementation of CI advice in real-world settings is a further area requiring investigation. Adherence has been identified as a significant confounder in multiple carbohydrate-related gestational diabetes studies [6]. One observational study, for instance, found that only 31% of meals adhered to the recommended CI range of 30–50 g, with 33% falling below and 34% exceeding this, such that no significant associations between target meal CI and glycaemic outcomes were observed [17].

One notable implementation concern is the excessive restriction of CI by some individuals attempting to normalise glucose levels, avoid insulin therapy, and prevent infant macrosomia [18]. The prevalence of this behaviour is uncertain; however, potential risks include nutritional inadequacy, maternal weight loss, starvation ketosis, the onset or exacerbation of disordered eating (for example, binge/restrict intake patterns and associated glycaemic variability), other adverse maternal mental health outcomes, and the risk of small-for-gestational-age infants [8,19]. Another concern is the potential for maternal hypoglycaemia if the individual starts insulin treatment, as this is associated with adverse neonatal outcomes [20].

Furthermore, the impact of self-monitored blood glucose (SMBG) on dietary intake and adherence remains unclear. While SMBG is an established component of gestational diabetes management [21], few studies have investigated its relationship with CI behaviours on a prospective basis. One observational study used an online dietary recall tool to investigate relationships between SMBG and intake frequency, and distribution and content. The study found that in-target fasting glycaemia was associated with the intake of three meals and three regular snacks per day, as compared with fewer eating episodes [17].

Figure 1 summarises key CI considerations and the current evidence base as outlined above.

With the global prevalence of gestational diabetes on the rise [19], further research is needed to explore CI behaviours in individuals with gestational diabetes, particularly in relation to SMBG in real-world clinical settings.

This study had two primary aims: (1) to describe the CI of individuals recently diagnosed with gestational diabetes, including the reasons behind low CI behaviours, and (2) to examine associations between CI and SMBG, within the context of broader nutritional and lifestyle factors.

## 2. Materials and Methods

### 2.1. Study Setting

This prospective observational study took place at a metropolitan teaching hospital in Sydney, Australia, located in a culturally and linguistically diverse region where over one in six pregnancies are affected by gestational diabetes. All pregnant individuals attending the hospital’s antenatal service underwent universal screening for gestational diabetes using a 75 g oral glucose tolerance test (OGTT) and the 2013 WHO diagnostic criteria [22]. Gestational diabetes was diagnosed if the fasting glucose level was >5.0 mmol/L (92 mg/dL), or 1 h glucose level was ≥10.0 mmol/L (180 mg/dL), or 2 h glucose level was ≥8.5 mmol/L (153 mg/dL).

After diagnosis, English-speaking individuals participated in group education sessions on diabetes management. These sessions included dietary guidance provided by a clinical dietitian, and training in SMBG by a diabetes nurse educator (DNE). Simplified written materials were provided to support low health literacy, with practical advice on serving sizes and carbohydrate distribution rather than formal carbohydrate counting. Underlying dietetic targets included the following: (1) total CI > 175 g/day; (2) regular 2–3 hourly eating pattern during waking hours, comprising three meals (breakfast, lunch, dinner) and 3–4 snacks; (3) CI of 30–50 g per main meal and 15–30 g per snack; (4) overnight fasting time of 8–10 h without CI.

Participants maintained routine care diaries the week following group education. A blank diary is provided as Appendix A for this study (Appendix A). These were reviewed during individual consultations with a dietitian and DNE after 5–8 days. Diary entries recorded the following: (1) SMBG readings; (2) timing of meals and SMBG readings; (3) out-of-bed and into-bed times; (4) types and quantities of food consumed at each eating episode; (5) episodes of poor sleep (yes/no), stress/illness/pain (scale of 1–3), hunger (scale of 1–3), and physical activity (above/below average and duration). SMBG was performed four times daily: fasting and 2 h post-meal (breakfast, lunch, and dinner).

### 2.2. Recruitment and Sample Size

All individuals attending English-language group education were invited to participate in this study on an opt-out consent basis, as the research did not involve interventions or data collection beyond routine care.

Given the lack of prior research directly comparable to the present study, a formal power calculation could not be performed. The absence of relevant studies made it difficult to estimate an appropriate sample size based on effect sizes or statistical benchmarks. Consequently, a target sample size of 90–100 datasets was determined using a practical approach, considering available resources and the feasibility of carrying out the study.

### 2.3. Inclusion Criteria

Participants were eligible for inclusion if they had a singleton pregnancy, were beyond 13 weeks of gestation, were over 18 years of age, and were able to complete a food diary in English. Exclusion criteria included the use of diabetes medications, significant illness during the study period, and a diagnosis of pre-existing Type 1 or Type 2 diabetes. Complete datasets were excluded if routine care diaries contained fewer than five days of comprehensive SMBG and CI data.

### 2.4. Data Collection

Data for this study centred on participant intake the week immediately following group education and prior to individualised dietitian therapy.

Baseline data were extracted from participants’ electronic medical records and included the following: age, pre-pregnancy weight and body mass index (BMI), gestational age, parity, history of gestational diabetes, self-reported ethnicity, OGTT results, haemoglobin A1c measured within 2 weeks of diagnosis, and weight recorded before and after food diary completion.

Dietary and SMBG data were collected from participants’ self-reported diaries. In accordance with routine care, participants were not required to weigh individual food items but estimated portion sizes using visual comparisons with standard household measures, such as a 250 mL cup. During the dietitian and DNE review appointments, recorded SMBG readings were verified for timing and accuracy using the inbuilt memory function of the participants’ SMBG meters. Corrections were made to the diaries as necessary.

SMBG data were excluded from analysis if procedural errors were identified, including the following: (1) SMBG conducted more than 10 min before or after the 2 h mark following meal commencement; (2) fasting SMBG performed more than 20 min after waking, even if prior to eating; (3) additional food intake occurring more than 30 min after meal commencement and prior to the corresponding SMBG reading.

Data on episodes of poor sleep, stress/illness/pain (combined), hunger, and physical activity were additionally collected from the diaries.

As a secondary analysis, the first three days of dietary data (comprising the first two weekdays and one weekend day) were entered into a computerised dietary analysis program (FoodWorks, Version 9, Xyris Software) based on national Australian food composition tables [23]. This process generated additional data on energy, macronutrient, micronutrient, and food group intake, which were subsequently incorporated into the study spreadsheets. The following inbuilt core food groups were included: vegetables, fruits, dairy, grain (including breads, cereals, rice, pasta), and protein (including meats, eggs, nuts, and legumes).

At the dietitian review appointment following diary completion, carbohydrate amounts and types were clarified if there was insufficient detail for analysis. Reasons for low CI behaviours were identified by the dietitian, based on collaborative discussion with the participant, as per routine practice. These were recorded on standard forms under four predefined categories: (1) Unaware—knowledge deficit; (2) Unaware—short-term forgetfulness; (3) Aware—short-term food preferences and lifestyle factors; and (4) Aware—intentional to address blood glucose, health, or weight. The dietitians conducting these assessments were not investigators in this study.

### 2.5. Study Variable Definitions

SMBG readings were classified as above target if fasting readings exceeded 5.0 mmol/L (92 mg/dL) or if postprandial readings exceeded 6.7 mmol/L (120 mg/dL), in accordance with recommendations by the Australian Diabetes in Pregnancy Society (ADIPS) [24]. Clinically elevated glycaemia was defined as more than 20% of glucose readings above target, consistent with the threshold proposed by ADIPS for considering additional interventions (i.e., diabetes medications) after dietary measures have been optimised [24]. Participants were provided with either the Freestyle Lite^®^ (Abbott Diabetes Care) or Accu-Chek Nano^®^ (Roche Diagnostics) blood glucose meters, both of which met ISO accuracy standards [25].

Low-carbohydrate days were defined as a CI of less than 135 g, consistent with definitions in prior research [26]. Low-carbohydrate meals and snacks were classified as containing less than 30 g and 15 g of carbohydrate, respectively, while high-carbohydrate meals and snacks were defined as exceeding 50 g of carbohydrate, reflecting common targets used by practicing dietitians [16,17]. Meals were categorised as high GI if the majority of carbohydrate consumed was identified as high GI based on routine dietitian assessment. Extended daytime fasts were defined as no nutritional intake for over 3.5 h, and extended nighttime fasts as exceeding 12 h, based on local consensus. Clinically significant patterns were defined as occurring more than twice per week, aiming to capture habitual rather than exceptional behaviour.

### 2.6. Data Analysis

Data were de-identified and entered into spreadsheets by a trained nutrition and dietetics student not involved in the clinical care of participants. Manual carbohydrate analysis was performed for all 5–8 completed diary days, based on 15 g carbohydrate servings (rounded to the nearest 0.5 serving). This simulated routine dietetic carbohydrate assessment, given that computerised analyses are not feasible in most clinical settings. In contrast, the computerised analysis captured only the first three diary days. Spreadsheet formulas calculated average and elevated SMBG readings, carbohydrate servings per day and per eating episode, episodes of intake above and below target ranges, and the timing between eating episodes. Standard deviations were used to assess variability in CI, timing, and SMBG for each participant. Episodic data were adjusted to a standard 7-day period. All spreadsheet entries were verified by a specialist diabetes dietitian.

Statistical analysis was conducted using The Jamovi Project (Version 2.3.21) [Computer Software]. A significance level of *p* < 0.05 was applied. Descriptive statistics are presented as frequencies (percentages) for categorical variables and medians (interquartile ranges) for continuous variables, as the majority of data were expected to be non-parametric. The Mann–Whitney U test was used to assess associations between above-target and in-target glycaemia for continuous variables, while Chi-square tests were employed for categorical variables. The Wilcoxon signed-rank test was used to compare intra-individual differences in paired average SMBG readings following eating episodes of varying carbohydrate amounts and after different durations of overnight fasting.

### 2.7. Ethical Approval and STROBE-Nut Checklist

This study was granted ethical approval by the South Western Sydney Local Health District Human Research Ethics Committee (HREC/17/LPOOL/91) on 26 April 2017. To ensure transparency and completeness in reporting, the STrengthening the Reporting of OBservational studies in Epidemiology—Nutritional Epidemiology (STROBE-nut) guidelines were applied [27]. This checklist is available as a Appendix A.

## 3. Results

A total of 97 datasets were included in the analysis after 12 participants opted out of the study and 34 datasets were excluded. The exclusions were as follows: four participants were under 13 weeks of gestation, two were significantly unwell during the study period, two had multiple pregnancies, and 26 lacked sufficient carbohydrate and blood glucose data for analysis. Among the included participants, five completed 5–6 diary days, while 92 completed 7–8 days. The final sample represented more than 2400 SMBG readings and over 3000 eating episodes.

### 3.1. Descriptive Data

As shown in Table 1, 31% of participants presented with clinically in-target glycaemia. Of those with clinically elevated glycaemia, 49% had elevated fasting glycaemia (EFG), 54% had elevated postprandial glycaemia (EPG), and 31% had both EFG and EPG. EPG was more common after dinner (34%) and lunch (32%) than breakfast (21%). More than two SMBG procedure errors per 7 days were observed for 62% of participants, with late postprandial testing being the most common error (33%).

The median numbers of carbohydrate servings per meal and snack were consistent with dietetic targets (2.7 and 1.3 servings, respectively), with breakfast intake slightly lower (2.3 servings) compared to lunch and dinner (2.9 servings each). However, more than a quarter of participants (29%) reported a low total CI (<135 g) on more than two days per week. Carbohydrate variability (SD > 1.5 servings) was most pronounced at dinner (36%), followed by lunch (27%), breakfast (10%), and snacks (3–5%)

The most common CI behaviours deviating from dietetic targets were missed snacks (93%), followed by extended daytime fasts (90%), high-carbohydrate meals (81%), extended overnight fasts (80%), high-glycaemic index meals (60%), and low-carbohydrate meals (59%). The least common deviations were overnight fasts of less than 8 h (5%), missed meals (28%), and low-carbohydrate snacks (50%). Meal time variability (SD > 1.5 h) was greatest for breakfast (15%) and lowest for lunch (9%).

Only a minority of participants documented hunger (3%), stress/illness/pain (13%), and physical activity episodes (44%). It was noted that many participants who recorded these episodes did so only during the first one or two days, with subsequent days left incomplete. Additionally, some participants offered minimal information regarding the quantities of low-carbohydrate foods, such as meats and vegetables. In these cases, a typical serving size was entered for the computerised analysis.

### 3.2. Reasons for Low CI Behaviours

As shown in Table 2, reviewing dietitians attributed low CI behaviours primarily to short-term food preferences and lifestyle factors (72% of events), rather than to a lack of knowledge (7%) or short-term forgetfulness (9%). Intentional CI restriction for the purpose of managing blood glucose, health, or weight was identified in nearly 12% of cases. However, no significant differences were observed in the prevalence of this behaviour between individuals with elevated versus in-target glycaemia.

### 3.3. Intra-Individual Carbohydrate Associations with Acute (Next Test) Glycaemia

As shown in Table 3, average SMBG readings were modestly higher two hours after higher carbohydrate meals compared to lower carbohydrate meals on an intra-individual basis. This pattern was consistent across all meal periods and was most pronounced in the comparison between in-target and high-carbohydrate meals (MD 0.54–0.60 mmol/L, *p* < 0.001). Consumption of 30–50 g of high-GI carbohydrate resulted in slightly higher average SMBG readings (MD 0.14 mmol/L, *p* = 0.042) compared with 30–50 g of lower GI carbohydrate; however, an intake of more than 50 g of high-GI carbohydrate led to a more pronounced difference (MD 1.23 mmol/L, *p* < 0.001).

Average fasting SMBG readings were slightly lower following low-carbohydrate dinner meals compared to regular carbohydrate dinners (MD −0.14, *p* = 0.047). No significant associations were observed for fasting or post-breakfast SMBG readings following skipped or low-carbohydrate pre-bed snacks. Fasting SMBG readings were marginally lower following an overnight fast of more than 12 h compared to an 8–10 h fast (MD −0.01, *p* = 0.016), although no differences were observed in post-breakfast SMBG readings.

No significant associations with acute glycaemia were found for the carbohydrate loads at the two meals and two snack periods preceding the primary test meal.

### 3.4. Dietary Associations with Clinically Elevated Fasting Glycaemia

As shown in Table 4, in comparison to in-target fasting glycaemia (TFG), EFG was inversely associated with total CI (Mdn 10.8 vs. 12.5 serves per day, *p* = 0.006); days with CI exceeding 175 g (Mdn 1.75 vs. 3.50 per 7 days, *p* = 0.037); carbohydrate serves per meal (Mdn 2.55 vs. 2.84, *p* = 0.01), high-carbohydrate meals (Mdn 3.94 vs. 6.89 per 7 days, *p* = 0.016); CI per day based on computerised analysis (Mdn 188 g vs. 231 g, *p* = 0.025); and intake from the grains food group (Mdn 6.54 vs. 8.39 serves/day, *p* = 0.045). Positive associations for EFG included low-carbohydrate days (Mdn 1.75 vs. 0.00 episodes per 7 days; *p* < 0.001); missed meals (Mdn 1.36 vs. 0.00 episodes per 7 days, *p* = 0.007); missed snacks (Mdn 10.50 vs. 7.50 episodes per 7 days, *p* = 0.038); low-carbohydrate meals (Mdn 4.33 vs. 2.00 episodes per 7 days, *p* = 0.004), and variability in overnight fasting time (Mdn SD 1.88 vs. 1.26 h, *p* = 0.020) as compared with TFG.

In contrast to the lower average fasting SMBG readings observed directly following low-carbohydrate dinner meals (described in Section 3.3), a strong positive association was found between EFG and TFG for low-carbohydrate dinner meals over the 7-day period (Mdn 1.75 vs. 0.00 episodes per 7 days, *p* < 0.001). No associations were found between EFG and TFG for overnight fasting time, missed evening snacks, or low-carbohydrate pre-bed snacks.

Based on the three-day computerised analysis, individuals with EFG had a lower dietary energy intake from carbohydrates (Mdn 40% vs. 44%, *p* = 0.026) and a higher intake from dietary fat (Mdn 35% vs. 33%, *p* = 0.047) compared with TFG.

No significant association was observed between EFG and TFG for pre-pregnancy body mass index (*p* = 0.172); however, current weight was higher in individuals with EFG (Mdn 76 kg vs. 67 kg, *p* = 0.013). Despite this, CI per kg per day continued to show a negative association with EFG (Mdn 2.47 g vs. 3.16 g, *p* = 0.002).

No significant differences were found between EFG and TFG for other baseline characteristics, other than the OGTT fasting glucose level and haemoglobin A1c. The number of eating episodes per day, timing of eating episodes, daytime fasting periods, and variability in these factors were all non-significant, as were the intake of high-glycaemic index meals, energy, fat, protein, fibre, micronutrients, and non-grain core food groups. Additionally, episodes of poor sleep, hunger, stress/illness/pain, and physical activity were not significantly associated with EFG.

### 3.5. Dietary Associations with Clinically Elevated Postprandial Glycaemia

Unlike EFG, EPG was not significantly associated with total CI, carbohydrate serves per meal and snack, and high-carbohydrate loads or low-carbohydrate meals and snacks over the 7-day period, as compared with TPG.

As shown in Table 5, positive associations were observed for high-glycaemic index lunches (Mdn 1.6 vs. 0.9 per week, *p* = 0.026) and dinners (Mdn 0.9 vs. 0.0 per 7 days, *p* = 0.023) between EPG and TPG for the corresponding meal periods. Additionally, dinner EPG was associated with missed lunch meals compared with dinner TPG (Mdn 0.9 vs. 0.0 per 7 days, *p* = 0.002).

According to computerised analysis, dietary energy (Mdn 7651 kJ vs. 9074 kJ per day, *p* = 0.031), protein (Mdn 91 g vs. 109 g per day, *p* = 0.015), fat (Mdn 61 g vs. 84 g per day, *p* = 0.003), fat per kilogram of body weight (Mdn 0.86 g vs. 1.15 g per day, *p* = 0.007), and energy derived from fat (Mdn 33% vs. 35%, *p* = 0.047) were significantly lower for EPG compared with TPG. Given that some Asian cuisines are higher in carbohydrates and lower in fat and protein, EPG and TPG were further compared across separate and combined Asian and non-Asian ethnic groups; however, no significant differences were identified.

The intake of multiple micronutrients from food sources was significantly lower for EPG compared with TPG. However, the concurrent use of micronutrient supplements was not evaluated.

No significant associations between EPG and TPG were observed for the number or timing of eating episodes per day, fasting durations during the day or night, baseline characteristics (other than 1 and 2 h OGTT glucose levels), intake from core food groups, or episodes of poor sleep, hunger, stress, illness, pain, or physical activity.

## 4. Discussion

To the authors’ knowledge, this is the most comprehensive investigation into CI behaviours and their associations with concurrent glycaemia in gestational diabetes. Despite the inherent limitations associated with analysing real-world clinical data, this study uncovered a range of significant associations. EFG was linked to lower total CI, as well as behaviours such as skipping meals and snacks, consuming low-carbohydrate meals, low-carbohydrate days, and a lower intake from the grains food group. In contrast, EPG was not associated with total CI, or high and low-carbohydrate eating episodes. Instead, EPG associations were found with high-glycaemic index meals and lower intakes of energy, protein, fat, and a range of micronutrients from food sources.

### 4.1. Carbohydrate Quantities

Postprandial SMBG readings were modestly higher directly following higher versus lower carbohydrate meals on a within-individual basis. This finding aligns with an older study that showed a lower proportion of energy from carbohydrates was needed to achieve lower glycaemic targets one hour after meals [5].

However, these acute observations did not translate into positive associations between EFG or EPG and higher carbohydrate meals or total CI per day over the 7-day analysis.

One potential explanation for these discrepancies is that established dietary behaviours may have been influencing metabolic parameters on a chronic basis—such as by affecting overall insulin resistance or pancreatic function—which would not be apparent when observing acute blood glucose patterns. Participants with EFG, for instance, had a higher proportion of energy derived from dietary fat, which has been shown to increase circulating free fatty acids and exacerbate insulin resistance [6,18]. A 2016 randomised clinical trial, in which all meals were provided, reported higher fasting glucose levels in individuals with gestational diabetes following a lower-carbohydrate/higher-fat diet (40% and 45% of total energy, respectively) for 6–7 weeks, compared with a high-complex-carbohydrate/lower-fat diet (60% and 25% of total energy, respectively) [28]. In the present study, dietary intake prior to group education was not explored, so it was not possible to determine whether observed CI behaviours were occurring before diagnosis.

An alternative explanation for the findings may be that individuals restricted or under-reported their CI in response to elevated SMBG readings. If this was the primary driver, two points merit consideration. First, it is unclear why EPG was not associated with CI behaviours similar to, or even more than, EFG. Notably, no ethnic differences in glycaemia or CI behaviours were observed that might explain this discrepancy. Second, it raises the question of why this phenomenon was not more strongly identified during the routine dietitian assessments. Instead, reviewing dietitians attributed low CI behaviours primarily to short-term food preferences and lifestyle factors. It is possible, however, that participants felt uncomfortable disclosing restrictive eating and under-reporting to their dietitian, or were not consciously aware of these behaviours.

Previous studies have shown higher rates of dietary under-reporting among women with higher body mass index (BMI), including during pregnancy [29]. However, the current study found no significant associations between pre-pregnancy BMI and either EFG or EPG, despite EFG participants having a higher weight than those with TFG.

Furthermore, the association between lower total CI and EFG was evident within the first three days of SMBG using computerised analysis. At this stage, participants had only 1–3 fasting SMBG readings recorded, effectively ruling out a longer-standing experience of glycaemia as the cause of low CI behaviours.

### 4.2. Carbohydrate Types

High-GI meals were positively associated with EPG (but not EFG) in both the intra-individual and 7-day analyses. Notably, the intra-individual analysis demonstrated that this association was most pronounced when high-GI meals were paired with a high-carbohydrate load (>50 g). For a moderate carbohydrate load (30–50 g), the association, though statistically significant, may not hold clinical relevance (MD 0.14 mmol/L).

This finding aligns with a recent meta-analysis which concluded that a low-glycaemic load (GL) diet reduced the risk of foetal macrosomia, whereas low-GI and low-carbohydrate diets produced outcomes comparable to standard care [11]. The beneficial effects of lower-GI and GL diets may stem from direct interactions with insulin and glucose metabolism or indirect impacts on overall dietary quality, body weight, and composition [9].

In the present study, no association was observed between fibre intake and EPG, although a recent systematic review and meta-analysis concluded that higher fibre diets can improve glycaemic control in pregnancies affected by gestational diabetes and Type 2 diabetes [30].

### 4.3. CI Timing, Distribution, and Variability

Other than an association between dinner EPG and skipped lunches, no clinically significant associations with EPG were found for missed meals and snacks, low-carbohydrate meals and snacks, number or timing of eating episodes, night and daytime fasting periods, variability in timing and CI, and hunger episodes. This was in contrast to EFG, which was associated with missed meals and snacks, as well as low-carbohydrate meals. This may suggest that a range of intake patterns could feasibly achieve target postprandial glycaemia and should be individualised.

Two hypotheses not supported by this study were (1) that more frequent eating episodes would lead to improved glycaemia, for example, through the prevention of hunger and overeating in response [31]; and (2) that extended overnight fasts and missed or low-carbohydrate bedtime snacks would be positively associated with EFG and/or EPG at breakfast, secondary to metabolic effects [32]. Other studies in gestational diabetes have similarly failed to find evidence to support these ideas. One observational study based on 24 h recalls found that longer overnight fasting intervals and fewer eating episodes per day were associated with improved fasting and 2 h glycaemia, respectively [33]. A randomised crossover study reported that a bedtime snack led to slightly higher fasting glycaemia [34]. A third study noted that increased CI from 6 pm until bedtime was incrementally associated with overall suboptimal glycaemia and higher birthweight neonates [35]. In contrast, one observational study found a positive association between EFG and a lower proportion of energy from carbohydrate at evening snacks [17]. In the present study, variability in overnight fasting time was positively associated with EFG and there was a marginal reduction in average fasting SMBG readings following extended overnight fasts of >12 h, compared with 8–10 h (MD −0.01 mmol/L). This was statistically, but not clinically, significant. Additionally, no associations were found between breakfast EPG and overnight fasting duration or CI at the previous dinner or evening snack period.

### 4.4. Other Dietary Components

EPG was significantly associated with lower intakes of energy, protein, fat, and a range of micronutrients from food sources based on the computerised analysis. If this reflected chronic intake, it may be speculated that lower-quality diets affected pancreatic insulin synthesis and release in response to carbohydrate loads, as no similar multi-nutrient associations were observed with EFG. For instance, dietary protein is a potent stimulator of insulin secretion [36], while numerous micronutrients play key roles in insulin synthesis, storage, and release [37]. Furthermore, dietary fat delays gastric emptying, which in turn slows carbohydrate absorption and may attenuate the postprandial glycaemic response [38].

A restriction in overall food intake in response to elevated SMBG readings may provide an alternative explanation. However, a concurrent reduction in CI intake would have been expected if this were the case, yet was not observed for EPG. Qualitative research has highlighted various psychological and behavioural impacts of a gestational diabetes diagnosis, including extreme behaviours in some individuals, such as purging and self-starvation due to fear for their baby [39]. However, there remains a paucity of research investigating the prevalence of such responses.

### 4.5. Additional Meal-Specific Associations

Elevated glycaemia was less common following breakfast meals than after lunch and dinner meals, despite three-quarters of participants having an average breakfast CI exceeding 30 g. This contrasts with the literature which suggests that CI should be restricted to 15–30 g at breakfast to prevent hyperglycaemia resulting from the morning peak in cortisol levels [40]. A randomised trial employing continuous glucose monitoring found mixed associations between glycaemia and higher versus lower morning CI. Consuming 50% of daily CI in the morning resulted in lower average and fasting glucose levels, as well as reduced insulin resistance, but was associated with higher glucose variability compared to consuming 50% of CI in the evening [13].

In the current study, dinner meals exhibited the highest variability in both glycaemia and CI. This may reflect that dinner is typically the main cooked meal of the day in Australia, offering the greatest variety of options for many individuals [41]. Skipped lunch meals were associated with dinner EPG, potentially due to increased levels of unreported hunger and higher compensatory eating at dinner. Previous research in non-pregnant individuals has demonstrated increased food consumption at the meal following a skipped meal, accompanied by a decline in overall dietary quality [42].

The association between EFG and low-carbohydrate meals was strongest for dinner meals based on the 7-day analysis (*p* < 0.001). However, average fasting SMBG readings were slightly lower the morning following low-carbohydrate dinner meals on an intra-individual basis. This discrepancy may indicate either a metabolic effect of habitual low-carbohydrate dinners or undisclosed dinner avoidance in response to prior EFG. If the latter were the case, similar associations might be expected for low-carbohydrate and/or skipped evening snacks, which were not observed. However, dietitian assessment identified intentional CI restriction as more prevalent for low-carbohydrate meals than for skipped or low-carbohydrate snacks. This may indicate that the type and size of the eating episode may influence intentional low CI behaviours.

### 4.6. Difference in Dietary Profiles for EFG and EPG

The distinction identified in this study between EFG and EPG may align with emerging recognition of gestational diabetes subtypes, most notably insulin-resistant (IRGD) versus insulin-deficient (IDGD) subtypes. IRGD is characterised by a higher BMI, elevated fasting glycaemia, and increased rates of adverse pregnancy outcomes compared to IDGD, which is typically observed in leaner individuals presenting with elevated postprandial glycaemia [43,44,45]. In this study, individuals with EFG had a higher weight compared to those with TFG, although BMI was not significantly different. They also consumed a lower proportion of energy from carbohydrates and a higher proportion of energy from fat, which has been associated with increased insulin resistance [13,28]. Conversely, participants with EPG may have consumed a lower quality diet, based on the multi-nutrient associations described earlier. In future, it may become evident that optimal dietary profiles differ between these and other gestational diabetes subtypes. If the associations observed in this study were driven by metabolic impacts of chronic nutrition—rather than the glycaemia driving the nutrition—this would suggest that some individuals with IRGD may benefit from a higher carbohydrate and lower fat intake, whereas those with IDGD might benefit from general (multi-nutrient) improvements in dietary quality.

### 4.7. Adherence to Carbohydrate Recommendations

This study highlighted difficulties experienced by newly diagnosed individuals in adhering to dietetic CI targets during the first week of active gestational diabetes management. More than 70% of participants recorded missed snacks, high-carbohydrate loads, and extended fasting periods on more than two days a week. Despite this, high-carbohydrate loads and extended fasting periods were not associated with EFG or EPG, challenging the appropriateness of discouraging these on a universal basis.

Moreover, dietitians identified short-term food preferences and lifestyle factors as more influential drivers of low CI behaviours than knowledge deficits. This aligns with the literature recognising that health knowledge alone is often insufficient to drive behavioural change [46]. An over-reliance on education as the primary catalyst for gestational diabetes management is particularly problematic, in light of psycho-social barriers that often hinder implementation [47].

### 4.8. Strengths and Limitations

The ‘real-life’ clinical setting represented both a strength and a limitation of this observational study. One strength is the reflection of the volume of considerations faced by routine-care clinicians as they interpret available data to effectively support individuals with gestational diabetes. Other strengths included the diverse, multicultural participant sample, the intensity of glycaemic monitoring, and the comprehensive capture of CI behaviours—including amounts, types, and timing of CI—alongside both intra- and inter-individual analyses on a prospective basis. These characteristics increase the likelihood that the findings can be generalised to other gestational diabetes settings. Additionally, the inclusion of multi-nutrient computerised analysis and the identification of factors contributing to suboptimal CI intake provided valuable insights, placing CI behaviours within a broader context.

Conversely, the use of a single site for recruitment, rather than multiple sites, and the exclusion of participants due to insufficient data or lack of proficiency in English, may have introduced a degree of sampling bias and affected the ability to generalise the findings. Moreover, the limited data collection period—immediately following group education and prior to individualised dietitian therapy—may not have been representative of prior or subsequent dietary intake and associated CI behaviours in the study participants. Additionally, the study was not adequately powered to investigate associations with clinical and birth outcomes, which represent the longer-term objective of optimising glycaemia.

Similar to clinical practice, this study was subject to potential data errors associated with SMBG procedures and devices, as well as the possibility of fictitious, inaccurate, or under-reported dietary intake. Routine dietitian assessments, including investigations into the reasons behind reported intakes, are vulnerable to interviewer bias and suboptimal disclosure by participants. The accuracy of both manual and computerised dietary analyses is influenced by multiple assumptions, compounded by the challenge of blinding such data during analysis.

Due to potential confounders, the significant findings of this study should be interpreted with caution, particularly regarding the non-carbohydrate dietary variables and the computerised analyses, which only reflected the first three days of intake. Unlike the manual CI analyses, it was not possible to examine other nutrients on an intra-individual basis. In particular, micronutrient intake remains unclear, as the concurrent use of supplements was not investigated.

Given the large volume of available data, the possibility of Type 1 statistical errors exists. Despite this, a more detailed analysis was deemed worthwhile, given the paucity research on many of the variables included, and the recognition that the purpose in identifying significant findings would be to inform future research directions.

Type 2 statistical errors were also likely, as some variables may have been underpowered due to the sample size. For example, many participants did not consistently record data on hunger, physical activity, and stress/illness/pain episodes, leading to non-significant and speculative associations between these factors, dietary intake, and glycaemia. This suggests that there are inherent limitations in the types and volume of data that can realistically be collected during routine clinical care.

### 4.9. Implications for Practice

Based on the findings of the present study, and acknowledging the unclear causality of the observed associations, it would seem prudent to continue advocating for dietary balance—avoiding low CI behaviours, while ensuring adequate energy, protein, fibre, and unsaturated fats, alongside sources of carbohydrates with a lower glycaemic index and high nutrient density.

Some of the local targets employed in this study may have been overly stringent, such as the 8–10 h overnight fasting period, which 95% of participants failed to meet on more than two out of seven days. Similarly, the target range of 30–50 g of carbohydrates per meal made it challenging to meet the overall carbohydrate target of more than 175 g per day, particularly when participants missed snacks, which occurred frequently.

Furthermore, the stringency of the glycaemic treatment targets recommended by the ADIPS [23] is also questionable, as these remain largely undefined at an international level [19]. Potentially, higher targets may promote a more balanced dietary intake and enhance metabolic parameters, such as insulin resistance, in certain individuals. For example, for those who meet postprandial glycaemic targets yet struggle to consume snacks, well-balanced meals containing up to 60–70 g of higher-quality carbohydrates may be reasonable.

In contrast, individuals presenting with highly problematic glycaemia could be offered more extensive medical nutrition therapy to address any barriers to dietary implementation. This could include not only individualised nutritional advice, but also health coaching strategies such motivational interviewing, which may improve readiness for lifestyle improvements [48,49].

Furthermore, given the discrepancy observed in this study between the intra-individual and 7-day glycaemic associations, concerns should be raised regarding reactive approaches to SMBG readings by both clinicians and consumers. In particular, potential adverse effects of intensive SMBG should be considered, including any tendency for glycaemia to overshadow other important objectives, such as ensuring nutritional adequacy, preventing excessive maternal weight gain, and avoiding disordered eating behaviours [19,50].

## 5. Conclusions

This observational study identified contrasting CI profiles for EFG compared to EPG in individuals with recently diagnosed gestational diabetes. EFG was associated with lower CI behaviours, including a lower total CI, missed snacks, low-carbohydrate meals, days with a CI < 135 g, and a higher fat intake. In contrast, EPG was not significantly associated with total CI, or low or high-carbohydrate meals and snacks. Instead, EPG was associated with high-glycaemic index meals and a lower intake of energy, protein, fat, and micronutrients These findings suggest that tailored dietary recommendations may be required for different glycaemic presentations, although the direction of the observed associations remains unclear.

A key purpose of this study was to highlight findings that could inform future research in more controlled and interventional settings. Many unanswered questions remain, including the generalisation of the associations to other gestational diabetes cohorts, whether CI behaviours change as pregnancy and treatments progress, and how best to support individuals in implementing effective dietary advice. Future studies could include the following: (1) randomised controlled trials investigating higher intakes of nutrient-dense carbohydrates alongside lower intakes of dietary fat in individuals with IRGD, particularly those presenting with EFG but not EPG; (2) observational studies examining the dietary intakes of individuals with different subtypes of gestational diabetes over time, including pre- and post-individualised dietetic therapy; (3) formal qualitative studies exploring the reasons behind suboptimal adherence to dietary recommendations, including interviews conducted by researchers who are not part of the treating team.

Such research is important, as current gestational diabetes management strategies are time- and labour-intensive for both consumers and clinicians. In light of the ongoing global rise in gestational diabetes prevalence, the optimisation of these strategies is pressing.

## Figures and Tables

**Figure 1 nutrients-17-00400-f001:**
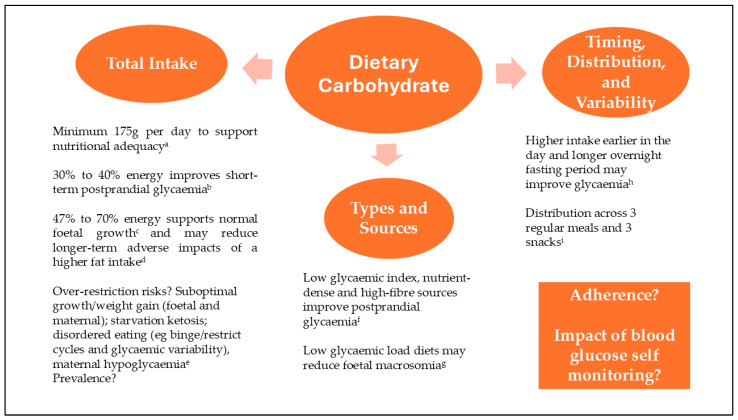
Carbohydrate intake considerations in gestational diabetes based on current evidence. ^a^ [15], ^b^ [5], ^c^ [8], ^d^ [7], ^e^ [8,9,20], ^f^ [7,9,10], ^g^ [11], ^h^ [12,13], ^i^ [16,17].

**Table 1 nutrients-17-00400-t001:** Baseline, blood glucose, and dietary intake descriptors during the first week of gestational diabetes self-management [*n* = 97; median (IQR) or *n* (%)].

Baseline characteristics		Carbohydrate variability	SD ^e^ >1.5 serves
Age (years) Gestational age (weeks) Pre-pregnancy BMI (kg/m^2^) OGTT diagnosis ^a^ - fasting alone - post-glucose load alone ^b^	32 (28–36) 28 (19–30) 25.10 (22.8–28.7) 24 (25.74%) 41 (42.28%)	- Breakfast - Lunch - Dinner - Morning snacks - Afternoon snacks - Evening snacks	10 (10.31%) 26 (26.80%) 35 (36.08%) 5 (5.15%) 5 (5.15%) 3 (3.09%)
- fasting and post-glucose load ^b^	32 (32.99%)	Carbohydrate intake behaviours	>2 episodes per week
Ethnicity self reported - South Asia - South East Asia - Middle East - Anglo-Australian - Other	29 (29.90%) 20 (20.62%) 19 (19.59%) 14 (14.43%) 15 (15.47%)	Carbohydrate < 135 g/day Missed meals - Breakfast - Lunch - Dinner Missed snacks	28 (28.87%) 27 (27.84%) 5 (5.15%) 11 (11.34%) 17 (17.53%) 90 (92.78%)
Blood glucose (BG) levels		- Morning	74 (76.29%)
In target (≤20% elevated ^c^) ^d^ Above target (>20% elevated) - Fasting - Postprandial (PP) combined ^e^ - Post-Breakfast - Post-Lunch - Post-Dinner - Both fasting and PP Procedure errors (>2 per week): - Late PP test (>130 min)	30 (30.93%) 48 (49.48%) 52 (53.61%) 20 (20.83%) 31 (32.29%) 33 (34.38%) 30 (30.93%) 60 (61.86%) 32 (32.99%)	- Afternoon - Evening Low-carbohydrate meals (<30 g) - Breakfast - Lunch - Dinner Low-carbohydrate snacks (<15 g) - Morning - Afternoon - Evening	43 (44.33%) 82 (84.54%) 57 (58.76%) 25 (25.77%) 17 (17.53%) 25 (25.77%) 48 (49.88%) 12 (12.37%) 17 (17.53%) 4 (4.12%)
Manual dietary analysis		High-carbohydrate loads (>50 g)	79 (81.44%)
Eating episodes per day Mealtime variability (SD ^f^ >1.5 h) - Breakfast - Lunch - Dinner Carbohydrate serves ^g^/day Carbohydrate serves/meal - Breakfast	5.43 (4.78–6.11) 14 (14.58%) 9 (9.38%) 11 (11.46%) 11.43 (9.88–13.29) 2.74 (2.29–3.19) 2.31 (2.00–2.94)	- Breakfast - Lunch - Dinner - Snacks High-glycaemic index meal Short overnight fast (<8 h) Extended overnight fast (>12 h) Daytime fast (>3.5 h)	20 (20.62%) 42 (43.30%) 84 (86.60%) 36 (37.11%) 58 (59.79%) 5 (5.15%) 76 (80.00%) 87 (89.69%)
- Lunch	2.86 (2.39–3.38)	Non dietary factors	>2 episodes per week
- Dinner Carbohydrate serves/snack	2.88 (2.25–3.50) 1.28 (1.10–1.55)	Hunger: any severity Physical activity above average Stress/illness/pain Poor sleep	3 (3.09%) 43 (44.33%) 13 (13.40%) 9 (9.28%)

^a^ 75 g oral glucose tolerance test (OGTT) diagnostic criteria: 0 h > 5.0, 1 h > 9.9, 2 h > 8.4 mmol/L. ^b^ Diagnosed on one and/or two-hour OGTT level; ^c^ Elevated: fasting > 5.0, 2 h PP > 6.7 mmol/L. ^d^ <20% fasting and <20% elevated at each PP meal period. ^e^ >20% elevated at one or more PP meal periods. ^f^ Standard deviation for the individual. ^g^ One serve = 15 g carbohydrate. A Appendix A is available for the following additional data: BMI categories, nulliparity, previous history of gestational diabetes, haemoglobin A1c, BG variability, BG procedure errors, computerised dietary analysis totals, weight change over the week.

**Table 2 nutrients-17-00400-t002:** Reasons for low-carbohydrate intake behaviours as per routine dietitian assessment (*n* = 97).

Non-Adherence Events	Unaware: Knowledge Deficit	Unaware: Short-Term Forgetfulness	Aware: Short-Term Food Preferences and Lifestyle Factors	Aware: Intentional to Address Blood Glucose, Health, or Weight
Missed meals	0	1	13	0
Missed snacks	1	4	74	4
Day fasts > 3.5 h	1	2	47	3
Overnight fasts > 10 h	1	10	31	3
Low-carbohydrate meals	11	8	36	20
Low-carbohydrate snacks	7	4	26	7
TOTAL	21 (6.69%)	29 (9.24%)	227 (72.29%)	37 (11.78%)
Participants with at least 1 event - EFG - TFG - EPG - TPG	9 (9.27%)	19 (19.59%)	84 (86.59%)	26 (26.80%) 14/48 (29.17%) 12/49 (24.49%) ^a^ 11/52 (21.15%) 15/45 (33.33%) ^b^

^a^ Elevated (EFG) vs. in-target fasting glycaemia (TFG) *p* = 0.603. ^b^ Elevated (EPG) vs. in-target postprandial glycaemia (TPG) *p* = 0.177.

**Table 3 nutrients-17-00400-t003:** Intra-individual comparisons of average blood glucose (BG) levels directly following selected dietary intake variables.

Comparison	Number with Pairwise Data	Mean Difference (95% CI)	*p* Value
Fasting BG average ^a^ (mmol/L) after			
- overnight fast > 12 h vs. 8–10 h	44	−0.01 (−0.23 to −0.03)	0.016 *
- low vs. in-target carbohydrate (carb.) dinner ^b^	55	−0.14 (−0.14 to 0.00)	0.047 *
- high vs. regular carb. dinner ^b^	71	0.02 (−0.03 to 0.08)	0.463
- (low-carb. or skipped) vs. regular pre-bed snack ^c^	69	−0.01 (−0.08 to 0.05)	0.850
Postprandial BG average ^a^ (mmol/L) after			
- overnight fast > 12 h vs. 8–10 h (breakfast)	37	0.10 (−0.22 to 0.41)	0.556
- low vs. in-target carb. meal ^b^			
- breakfast	42	−0.23 (−0.44 to 0.00)	0.052
- lunch	40	−0.50 (−0.78 to −0.23)	0.001 **
- dinner	52	−0.35 (−0.50 to −0.18)	0.001 **
- high vs. in-target carb. meal ^b^			
- breakfast	45	0.60 (0.33 to 0.90)	<0.001 ***
- lunch	56	0.56 (0.33 to 0.77)	<0.001 ***
- dinner	61	0.54 (0.31 to 0.77)	<0.001 ***
- high-glycaemic index (30–50 g) vs. in-target carb. meal	64	0.14 (0.01 to 0.28)	0.042 *
- high-glycaemic index (>50 g) vs. in-target carb. meal	30	1.23 (0.91 to 1.51)	<0.001 ***
- low/high vs. regular carb. for each of the two meals preceding the primary test meal			≥0.05
- low/high vs. regular carb. for each of the two snack periods preceding the primary test meal			≥0.05

^a^ Average for the individual. ^b^ Low carb: <30 g, in-target carb: 30–50 g, high carb: >50 g. ^c^ Low carb: <15 g, in-target carb: 15–30 g. * *p* < 0.05, ** *p* < 0.01, *** *p* < 0.001.

**Table 4 nutrients-17-00400-t004:** Dietary and other relevant associations for individuals with clinically elevated versus in-target fasting glycaemia.

Significant Associations (*p* < 0.05)	Clinically Elevated ^a^ Fasting Glycaemia *n* = 48, Median (IQR)	Clinically In-Target Fasting Glycaemia *n* = 49, Median (IQR)	*p* Value
Inverse:			
Carbohydrate (carb.) serves ^b^ per day	10.80 (9.11–12.67)	12.50 (10.36–14.13)	0.006 **
Carb. > 175 g per day (n/7 days)	1.75 (0.66–3.72)	3.50 (1.56–5.25)	0.037 *
Carb. serves per meal (average ^c^)	2.55 (2.20–2.96)	2.84 (2.48–3.35)	0.010 *
- Carb. serves per breakfast (average)	2.20 (1.88–2.82)	2.50 (2.17–2.94)	0.039 *
- Carb. serves per dinner (average)	2.50 (2.20–3.00)	3.14 (2.50–3.86)	0.002 **
High-carb. meals (>50 g) combined (n/7 days)	3.94 (2.41–6.13)	6.89 (3.11–8.75)	0.016 *
- High-carb. breakfasts (n/7 days)	0.39 (0.00–1.75)	0.88 (0.00–2.33)	0.042 *
- High-carb. dinners (n/7 days)	1.75 (0.88–2.08)	3.00 (1.00–4.38)	0.003 **
Total carb. (g/day computerised analysis)	188 (158–244)	231 (188–266)	0.025 *
Carb. per kg (g/day)	2.47 (1.98–3.22)	3.16 (2.45–4.09)	0.002 **
Energy from carb. (%)	40.37 (36.56–47.36)	43.59 (38.69–50.02)	0.026 *
Grains group (servings ^d^ per day)	6.54 (5.11–8.84)	8.39 (5.90–10.28)	0.045 *
Positive:			
Low-carb. days (<135 g) (n/7 days)	1.75 (0.58–3.13)	0.00 (0.00–1.75)	<0.001 ***
Missed meals combined (n/7 days)	1.36 (0.00–4.45)	0.00 (0.00–1.00)	0.007 **
- Missed lunches (n/7 days)	0.88 (0.00–1.17)	0.00 (0.00–0.88)	0.006 **
- Missed dinners (n/7 days)	0.00 (0.00–2.63)	0.00 (0.00–0.88)	0.038 *
Missed snacks combined (n/7 days)	10.50 (6.74 −15.06)	7.50 (6.00–10.89)	0.038 *
Low-carb. meals (<30 g) combined (n/7 days)	4.33 (1.84–6.34)	2.00 (1.00–4.38)	0.004 **
- Low-carb. dinners (n/7 days)	1.75 (0.88–2.67)	0.00 (0.00–1.00)	<0.001 ***
Energy from fat (%)	35.06 (31.76–39.20)	32.96 (28.01–37.40)	0.047 *
Variability in overnight fasting time (SD ^e^, h)	1.88 (1.39–2.74)	1.26 (1.3–2.15)	0.020 *
OGTT ^f^ fasting glucose level (mmol/L)	5.1 (4.9–5.3)	4.8 (4.4–5.2)	0.004 **
Haemoglobin A1 c (%)	5.2 (5.0–5.4)	5.0 (4.8–5.1)	0.003 **
Pre-pregnancy weight (kg)	65 (60–83)	61 (54–69)	0.032 *
Current weight (kg)	75.7 (67.99–87.01)	67.2 (59.29–78.20)	0.013 *
Non-significant Associations (*p* ≥ 0.05) ^g^ Age; pre-pregnancy BMI (*p* = 0.172), OGTT ^f^ 1 and 2 h glucose levels; gestational age; previous gestational diabetes; nulliparity; ethnicity; postprandial blood glucose elevations; blood glucose procedure errors; weight change in the week. Number of eating episodes per day; carbohydrate serves per lunch and per snack; missed breakfasts and separated snack times; low-carbohydrate breakfasts, lunches, and snacks; high-carbohydrate lunches and snacks; high-glycaemic index meals; variability in timing of eating episodes; overnight fasting time < 8 h, >10 h and >12 h; day fasts > 3.5 h. Computerised analysis: energy, fat, protein, and fibre intake; % energy from protein; micronutrient intakes from food sources; intake from the vegetables, fruit, dairy, and protein food groups. Episodes of poor sleep, hunger, stress/illness/pain, and physical activity.			

^a^ More than 20% of readings > 5.0 mmol/L. ^b^ One serve = 15 g carbohydrate. ^c^ Average for the individual. ^d^ Serve sizes as per FoodWorks (Version 9, Xyris Software). ^e^ SD for the individual. ^f^ 75 g oral glucose tolerance test. ^g^ Detail not shown due to volume; a Appendix A is available. * *p* < 0.05, ** *p* < 0.01, *** *p* < 0.001.

**Table 5 nutrients-17-00400-t005:** Dietary and other relevant associations for individuals with clinically elevated versus in-target postprandial glycaemia.

Significant Associations ^a^ (*p* < 0.05)	Clinically Elevated ^b^ Postprandial Glycaemia *n* = 52, Median (IQR)	Clinically in-Target Postprandial Glycaemia *n* = 45, Median (IQR)	*p* Value
Inverse			
Energy intake (kJ/day)	7651 (6602–9488)	9074 (7473–10,714)	0.031 *
Protein intake (g/day)	91 (76–105)	109 (83–126)	0.015 *
Fat intake (g/day)	61 (49–84)	84 (69–107)	0.003 **
- Fat per kg (g/day)	0.86 (0.69–1.16)	1.15 (0.84–1.27)	0.007 **
- Energy from fat (%)	32.57 (27.50–38.23)	34.74 (39.36)	0.047 *
Micronutrients from food sources (per day)			
- Vitamin A equivalents (mg)	625 (421–1066)	951 (619–1442)	0.003 **
- Niacin equivalents (mg)	38.63 (31.30–45.38)	46.10 (34.23–53.58)	0.044 *
- Potassium (mg)	2564 (2179–3289)	2915 (2595–3725)	0.021 *
- Magnesium (mg)	288 (207–360)	315 (280–379)	0.038 *
- Calcium (mg)	661 (531–928)	845 (627–1114)	0.034 *
- Phosphorus (mg)	1340 (1211–1584)	1577 (1310–1817)	0.018 *
- Iron (mg)	9.08 (7.20–12.07)	11.32 (8.53–15.14)	0.005 **
- Zinc (mg)	9.82 (8.04–12.08)	12.09 (9.16–14.85)	0.009 **
- Vitamin E (mg)	9.83 (7.33–13.34)	13.08 (10.52–16.91)	0.004 **
- Vitamin B12 (μg)	3.59 (2.96–4.60)	4.73 (3.45–6.04)	0.004 **
- Food folate (μg)	284 (217–346)	370 (253–419)	0.006 **
- Selenium (μg)	79.73 (63.39–98.54)	97.44 (75.35–110.72)	0.045 *
Positive			
OGTT ^c^ 1 h level (mmol/L)	10.1 (9.2–10.9)	9.2 (7.3–10.6)	0.010 *
OGTT 2 h level (mmol/L)	8.5 (7.2–9.1)	6.5 (5.9–8.0)	<0.001 ***
Variability: postprandial BG (SD, mmol/L) ^d^	0.88 (0.79–1.03)	0.62 (0.54–0.71)	<0.001 ***
Meal-specific			
Lunch EPG ^e^ only:			
- high-glycaemic index lunches (n/7 days)	1.56 (0.88–2.63)	0.88 (0.00–1.75)	0.026 *
Dinner EPG only:			
- missed lunches (n/7 days)	0.88 (0.00–2.00)	0.00 (0.00–0.88)	0.002 **
- high-glycaemic index dinners (n/7 days)	0.88 (0.00–1.75)	0.00 (0.00–1.08)	0.023 *
Non-Significant (NS, *p* ≥ 0.05)			
Carb. serves ^f^ per day	11.38 (9.63–13.08)	11.50 (10.08–13.34)	0.664
Carb. serves per meal	2.76 (2.26–3.20)	2.74 (2.38–3.18)	0.960
Carb. grams per day (computerised analysis)	200 (174–263)	221 (176–249)	0.663
- Carb. per kg (g per day)	2.85 (2.31–3.60)	3.05 (2.17–3.70)	0.919
- Energy from carbohydrate (%)	43.08 (38.90–49.42)	41.04 (37.38–45.77)	0.079
Other NS ^g^: Maternal age; pre-pregnancy weight, pre-pregnancy BMI, OGTT fasting glucose level; haemoglobin A1c, gestational age; previous gestational diabetes; nulliparity; ethnicity; Asian ethnicity combined; fasting blood glucose elevations; blood glucose procedure errors; current weight; weight change in the week. Number of eating episodes per day; <9 and >12 serves per day; carb. serves per snack; missed breakfasts, dinners, and snacks; low-carb. meals and snacks; high-carb. meals and snacks; variability in timing of eating episodes; variability in carb. at meals; overnight fasting time; day fasts > 3.5 h. Computerised analysis: energy per kg; protein per kg; % energy from protein; dietary fibre; intake from fruit, grains, vegetables, protein, and dairy food groups. Episodes of poor sleep, hunger, stress/illness/pain, and physical activity.			

^a^ All meals combined unless otherwise stated. ^b^ More than 20% of readings > 6.7 mmol/L. ^c^ 75 g oral glucose tolerance test. ^d^ SD in blood glucose for the individual. ^e^ EPG: elevated postprandial glycaemia. ^f^ Carbohydrate serves; one serve = 15 g. ^g^ Detail not shown due to volume; a Appendix A is available. * *p* < 0.05, ** *p* < 0.01, *** *p* < 0.001.

## Data Availability

The data that support the findings of this study are available from the corresponding author upon reasonable request.

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
