# Peer review of "Associations Between Carbohydrate Intake Behaviours and Glycaemia in Gestational Diabetes: A Prospective Observational Study"

_nutrients, 2025, doi:10.3390/nu17030400_

Round 1

Reviewer 1 Report

Comments and Suggestions for Authors

To the Authors

The Authors of the Ms ID nutrients-3441977 describe carbohydrate intake (CI) behaviours in individuals recently diagnosed with gestational diabetes Gestational diabetes mellitus (GDM) and identify associations with self monitored blood glucose (SMBG) levels in real-life clinical settings. This observational study identified contrasting CI profiles for EFG compared to EPG in individuals with recently diagnosed GDM, with notable associations between lower CI behaviours and EFG, as well as the absence of a positive association between CI and EPG (elevated postprandial glycaemia). These findings suggest that,  beyond universal recommendations and short- term blood glucose monitoring, there is a need for tailored dietary guidelines. The study is well written, and its rationale well explained. Methods have been carefully planned and accurately described. Strengths and weaknesses of the present study are accurately explored. While I do agree with the general conclusions of the study and a need for further interventional trials in examining the impact of CI on EFG/TFG or EPG/TPG, I would suggest paying more attention to a few issues that are in need to be addressed.

Major Issues

1.    HbA1c data are not reported. Prior reports have indicated a prognostic role of this variable (e.g., Muhuza MPU, et al. The association between maternal HbA1c and adverse outcomes in gestational diabetes. Front Endocrinol (Lausanne). 2023 Mar 16; 14:1105899]. This should be included among the study limitations. Please clarify and/or justify.

2.    Clinical follow-up in terms of pregnancy adverse effects in the examined GDM population is missing. In particular, it would be interesting to know the relationship between pregnancy adverse outcomes [macrosomia, pregnancy-induced hypertension (PIH), preterm birth, and primary Cesarean section] and EFG/TFG or EPG/TPG. This should be included among the study limitations.  Please clarify and/or justify

Minor issues

1.     Abstr. L 30 “higher average glucose levels two hours after high-carbohydrate (>50 g) verses regular carbohydrate” à Should “verses” be read as “serves”? Please clarify.

2.    Given the relatively large number of abbreviations, I would suggest including an abbreviation list to the text for a better readability.

Author Response

Comment 1: The Authors of the Ms ID nutrients-3441977 describe carbohydrate intake (CI) behaviours in individuals recently diagnosed with gestational diabetes Gestational diabetes mellitus (GDM) and identify associations with self monitored blood glucose (SMBG) levels in real-life clinical settings. This observational study identified contrasting CI profiles for EFG compared to EPG in individuals with recently diagnosed GDM, with notable associations between lower CI behaviours and EFG, as well as the absence of a positive association between CI and EPG (elevated postprandial glycaemia). These findings suggest that,  beyond universal recommendations and short- term blood glucose monitoring, there is a need for tailored dietary guidelines. The study is well written, and its rationale well explained. Methods have been carefully planned and accurately described. Strengths and weaknesses of the present study are accurately explored. While I do agree with the general conclusions of the study and a need for further interventional trials in examining the impact of CI on EFG/TFG or EPG/TPG, I would suggest paying more attention to a few issues that are in need to be addressed.

Response 1: Thankyou for your constructive review and affirmation of the various components of this paper. We have attempted to address the issues raised in our responses below.

Comment 2: HbA1c data are not reported. Prior reports have indicated a prognostic role of this variable (e.g., Muhuza MPU, et al. The association between maternal HbA1c and adverse outcomes in gestational diabetes. Front Endocrinol (Lausanne). 2023 Mar 16; 14:1105899]. This should be included among the study limitations. Please clarify and/or justify.

Response 2: Thankyou for highlighting this point. We have now incorporated data on HbA1c that were available but had not previously been analysed. These data have been added to the methodology in Section 2.4 (paragraph 2) and referenced in the footnote of Table 1. HbA1c was found to be significantly associated with EFG (p = 0.003), as now presented in Table 4 and discussed in Section 3.4 (paragraph 5). For EPG, HbA1c showed a borderline non-significant association (p = 0.052, based on the Student’s t-test, as the data were parametric). These findings are now included in Supplementary Table 3 and the non-significant section of Table 5. Although these results do not alter the overall dietary findings or conclusions of this study, they provide further insight into the distinctions between the EFG and EPG presentations. This adds to the growing body of evidence supporting the existence of gestational diabetes subtypes. As this was not the primary focus of the study, an additional table summarising the associations between elevated glycaemia, oral glucose tolerance test (OGTT) results, HbA1c, pre-pregnancy weight, and body mass index has been included in the supplementary material (Supplementary Table 4).

Comment 3: Clinical follow-up in terms of pregnancy adverse effects in the examined GDM population is missing. In particular, it would be interesting to know the relationship between pregnancy adverse outcomes [macrosomia, pregnancy-induced hypertension (PIH), preterm birth, and primary Cesarean section] and EFG/TFG or EPG/TPG. This should be included among the study limitations.  Please clarify and/or justify

Response 3: Agreed. This has been clarified within the study limitations (section 4.8 paragraph 2). Due to resourcing limitations, the current study was not powered to capture differences in clinical and birth outcomes, which typically require hundreds to thousands of participants, depending on the outcome in question.  

Comment 4:  Abstr. L 30 “higher average glucose levels two hours after high-carbohydrate (>50 g) verses regular carbohydrate” à Should “verses” be read as “serves”? Please clarify.

Response 4: This section of the abstract has been completely removed in response to another review which recommended abbreviating the abstract.

Comment 5: Given the relatively large number of abbreviations, I would suggest including an abbreviation list to the text for a better readability.

Response 5: Agreed. A list of study abbreviations has been included prior to the introduction

Reviewer 2 Report

Comments and Suggestions for Authors

I would like to congratulate the authors for the carried out study. In my opinion, it deserves to be considered for publication after the following revisions:

The abstract is not adequate. According to the journal’s guidelines, it should be no more than 250 words.

It would be great if a table is included in the Introduction with the organized presented data.

Why did you only consider one center to conduct your research? This is a relevant limitation when you want to make your results representative of the study population. The sample size must also be properly justified.

In section 2.7 you need to include the study’s approval date.

The Results and Discussion sections are well presented. However, the results obtained should be better highlighted and clarified in the Conclusions.

Author Response

Comment 1: I would like to congratulate the authors for the carried out study. In my opinion, it deserves to be considered for publication after the following revisions:

Response 1: Thankyou for affirming this study and providing a helpful review. We have attempted to address your suggested revisions as outlined below.

Comment 2: The abstract is not adequate. According to the journal’s guidelines, it should be no more than 250 words.

Response 2: Agreed. The abstract has been significantly condensed, with its length reduced by nearly half through streamlined wording and the removal of references to secondary investigations: the intra-individual analysis and the exploration of reasons for low CI behaviours. These findings are more appropriately addressed in the discussion section, where they provide greater context for interpreting the primary results

Comment 3: It would be great if a table is included in the Introduction with the organized presented data.

Response 3: Thankyou for this idea. After attempting a table, a diagram has instead been inserted into the introduction that summarises key carbohydrate intake considerations based on the evidence-base presented in the text (see Figure 1).

Comment 4: Why did you only consider one center to conduct your research? This is a relevant limitation when you want to make your results representative of the study population.

Response 4: Agreed. We have now incorporated your point as a study limitation in Section 4.8 (paragraph 2). As government-employed clinicians, rather than academic or privately funded researchers, our capacity to co-ordinate and resource projects beyond our own site or centre is constrained. Nevertheless, our access to ‘real-life’ clinical data within a population characterised by high rates of socioeconomic disadvantage and low health literacy offers valuable insights that may not be as readily accessible to privately funded projects

Comment 5: The sample size must also be properly justified.

Response 5: Agreed. This has been more clearly explained in section 2.2

Comment 6: In section 2.7 you need to include the study’s approval date.

Response 6: Agreed. This has now been included.

Comment 7: The Results and Discussion sections are well presented. However, the results obtained should be better highlighted and clarified in the Conclusions.

Response 7: Agreed. The primarily findings are now detailed and clarified in paragraph 1 of the conclusion.